# Postmarketing active surveillance of myocarditis and pericarditis following vaccination with COVID-19 mRNA vaccines in persons aged 12 to 39 years in Italy: A multi-database, self-controlled case series study

Marco Massari[1⊙], Stefania Spila Alegiani[1⊙], Cristina Morciano[1]*, Matteo Spuri[2], Pasquale Marchione[3], Patrizia Felicetti[3], Valeria Belleudi[4], Francesca Romana Poggi[4], Marco Lazzeretti[5], Michele Ercolanoni[5], Elena Clagnan[6], Emanuela Bovo[7], Gianluca Trifirò[8], Ugo Moretti[8], Giuseppe Monaco[9], Olivia Leoni[9], Roberto Da Cas[1], Fiorella Petronzelli[3], Loriana Tartaglia[3], Nadia Mores[10], Giovanna Zanoni[11], Paola Rossi[12], Sarah Samez[13], Cristina Zappetti[12], Anna Rosa Marra[3], Francesca Menniti Ippolito[1], on behalf of the TheShinISS-Vax|COVID Surveillance Group[¶]

1 National Centre for Drug Research and Evaluation, Istituto Superiore di Sanità (National Institute of Health), Rome, Italy, 2 Department of Infectious Diseases, Istituto Superiore di Sanità (National Institute of Health), Rome, Italy, 3 Department of post-marketing surveillance, Agenzia Italiana del Farmaco (Italian Medicines Agency), Rome, Italy, 4 Department of Epidemiology ASL Roma 1, Lazio Regional Health Service, Rome, Italy, 5 Business Intelligence, Data Science e Data Analysis, ARIA S.p.A., Milan, Italy, 6 ARCS–Azienda Regionale di Coordinamento per la Salute, Udine, Italy, 7 Veneto Tumour Registry, Azienda Zero, Padova, Italy, 8 Department of Diagnostics and Public Health, University of Verona, Verona, Italy, 9 Department of Health of Lombardy Region, Epidemiology Observatory, Milan, Italy, 10 Institute of Pharmacology, Pharmacovigilance, Policlinico Universitario A. Gemelli, Catholic University of Sacred Heart, Rome, Italy, 11 Immunology Unit, University Hospital, Verona, Italy, 12 Direzione centrale salute, politiche sociali e disabilità, Friuli Venezia Giulia Region, Trieste, Italy, 13 Centro Regionale di Farmacovigilanza, Friuli Venezia Giulia Region, Trieste, Italy

⊙ These authors contributed equally to this work.
¶ The members of the group are listed in the Acknowledgments.
* cristina.morciano@iss.it

**Data Availability Statement:** Data cannot be shared publicly under article 9 of Regulation (EU)

## Abstract

### Background

Myocarditis and pericarditis following the Coronavirus Disease 2019 (COVID-19) mRNA vaccines administration have been reported, but their frequency is still uncertain in the younger population. This study investigated the association between Severe Acute Respiratory Syndrome Coronavirus 2 (SARS-CoV-2) mRNA vaccines, BNT162b2, and mRNA-1273 and myocarditis/pericarditis in the population of vaccinated persons aged 12 to 39 years in Italy.

### Methods and findings

We conducted a self-controlled case series study (SCCS) using national data on COVID-19 vaccination linked to emergency care/hospital discharge databases. The outcome was the

2016/679. Data are available from the Data Protection Officer of Istituto Superiore di Sanità-Dott. Carlo Villanacci, e-mail: responsabile.protezionedati@iss.it, for researchers who meet the criteria for access to confidential data.

**Funding:** The Istituto Superiore di Sanità received funding from AIFA (Italian Medicines Agency) www.aifa.gov.it for this study in the framework of the collaboration agreement "Efficacia real world e sicurezza dei vaccini anti Covid-19: studio di coorte e Self-Controlled Case Series" (Effectiveness and safety of COVID-19 vaccines: cohort and Self-Controlled Case Series studies). AIFA is the Italian national regulatory body for drugs and vaccines and a public organization. All authors, including authors affiliated with AIFA are independent from the funder. The funders had no role in study design, data collection and analysis, decision to publish, or preparation of the manuscript.

**Competing interests:** I have read the journal's policy and the author of this manuscript have the following competing interests: in the last 36 months, GT coordinated a pharmacoepi team at the University of Messina till Oct 2020 and currently at the academic spin-off INSPIRE that received research grants from PTC Therapeutics, Kiowa Kirin, Chiesi, Daiichi Sankyo for the conduct of observational studies on topics not related to the paper; GT participated to Advisory Board/interview sponsored by Eli Lilly, Amgen, Sanofi, SOBI, Gilead, ABBvie, Verpora and Daiichi Sankyo on topics not related to the paper.

**Abbreviations:** AIFA, Agenzia Italia del Farmaco (Italian Medicines Agency); CI, confidence interval; COVID-19, Coronavirus Disease 2019; EC, excess of case; EMA, European Medicines Agency; EU, European Union; HR, hazard ratio; ICD-9-CM, International Classification of Disease 9th revision Clinical Modification; IRR, incidence rate ratio; IQR, Interquartile range; MIS-C, Multi-system inflammatory syndrome in children; PRAC, Pharmacovigilance Risk Assessment Committee; RI, relative incidence; RMP, risk management plan; SARS-CoV-2, Severe Acute Respiratory Syndrome Coronavirus 2; SCCS, self-controlled case series; SPEAC, Safety Platform for Emergency vACcines; UK, United Kingdom; US, United States.

first diagnosis of myocarditis/pericarditis between 27 December 2020 and 30 September 2021. Exposure risk period (0 to 21 days from the vaccination day, subdivided in 3 equal intervals) for first and second dose was compared with baseline period. The SCCS model, adapted to event-dependent exposures, was fitted using unbiased estimating equations to estimate relative incidences (RIs) and excess of cases (EC) per 100,000 vaccinated by dose, age, sex, and vaccine product. Calendar period was included as time-varying confounder in the model. During the study period 2,861,809 persons aged 12 to 39 years received mRNA vaccines (2,405,759 BNT162b2; 456,050 mRNA-1273); 441 participants developed myocarditis/pericarditis (346 BNT162b2; 95 mRNA-1273). Within the 21-day risk interval, 114 myocarditis/pericarditis events occurred, the RI was 1.99 (1.30 to 3.05) after second dose of BNT162b2 and 2.22 (1.00 to 4.91) and 2.63 (1.21 to 5.71) after first and second dose of mRNA-1273. During the [0 to 7) days risk period, an increased risk of myocarditis/pericarditis was observed after first dose of mRNA-1273, with RI of 6.55 (2.73 to 15.72), and after second dose of BNT162b2 and mRNA-1273, with RIs of 3.39 (2.02 to 5.68) and 7.59 (3.26 to 17.65). The number of EC for second dose of mRNA-1273 was 5.5 per 100,000 vaccinated (3.0 to 7.9). The highest risk was observed in males, at [0 to 7) days after first and second dose of mRNA-1273 with RI of 12.28 (4.09 to 36.83) and RI of 11.91 (3.88 to 36.53); the number of EC after the second dose of mRNA-1273 was 8.8 (4.9 to 12.9). Among those aged 12 to 17 years, the RI was of 5.74 (1.52 to 21.72) after second dose of BNT162b2; for this age group, the number of events was insufficient for estimating RIs after mRNA-1273. Among those aged 18 to 29 years, the RIs were 7.58 (2.62 to 21.94) after first dose of mRNA-1273 and 4.02 (1.81 to 8.91) and 9.58 (3.32 to 27.58) after second dose of BNT162b2 and mRNA-1273; the numbers of EC were 3.4 (1.1 to 6.0) and 8.6 (4.4 to 12.6) after first and second dose of mRNA-1273. The main study limitations were that the outcome was not validated through review of clinical records, and there was an absence of information on the length of hospitalization and, thus, the severity of the outcome.

## Conclusions

This population-based study of about 3 millions of residents in Italy suggested that mRNA vaccines were associated with myocarditis/pericarditis in the population younger than 40 years. According to our results, increased risk of myocarditis/pericarditis was associated with the second dose of BNT162b2 and both doses of mRNA-1273. The highest risks were observed in males of 12 to 39 years and in males and females 18 to 29 years vaccinated with mRNA-1273. The public health implication of these findings should be considered in the light of the proven mRNA vaccine effectiveness in preventing serious COVID-19 disease and death.

## Author summary

### Why was this study done?

Pharmacovigilance reports and observational studies have suggested an increased risk of myocarditis/pericarditis following the Coronavirus Disease 2019 (COVID-19) mRNA vaccine administration in people younger than 40 years.

More information on the safety of COVID-19 mRNA vaccines is needed to further explore the relationship between mRNA vaccines and myocarditis/pericarditis in this population.

## What did the researchers do and find?

We conducted a multiregional self-controlled case series (SCCS) study in Italy between 27 December 2020 to 30 September 2021 to investigate the association between myocarditis/pericarditis and COVID-19 mRNA vaccines in the population aged 12 to 39 years ($n$ = 2,861,809).

We found 441 myocarditis/pericarditis cases, 114 of which occurred within the 21-day risk interval after vaccination.

Within the 21-day risk interval, the relative incidence (RI) was 1.99 (95% confidence interval [CI] 1.30 to 3.05) after the second dose of BNT162b2 and 2.22 (1.00 to 4.91) and 2.63 (1.21 to 5.71) after the first and second doses of mRNA-1273, respectively. Within the 0 to 7-day risk interval, the RI was 6.55 (2.73 to 15.72) after first dose of mRNA-1273 and 3.39 (2.02 to 5.68) and 7.59 (3.26 to 17.65) after the second doses of BNT162b2 and mRNA-1273, respectively.

The highest risk was seen in males, 0 to 7 days after the first and second dose of mRNA-1273 (RIs of 12.28 (4.09 to 36.83) and 11.91 (3.88 to 36.53), respectively). After the second dose of mRNA-1273 in males, the excess of cases (EC) was 8.8 (4.9 to 12.9) per 100,000 vaccinated individuals.

## What do these findings mean?

Consistent with previous studies, the findings suggest that COVID-19 mRNA vaccines were associated with myocarditis/pericarditis in the population younger than 40 years. The results provide information that could be helpful for the continuous assessment of the postmarketing benefit/risk profile of the COVID-19 mRNA vaccines and should be considered within the context of the proven mRNA vaccine effectiveness in reducing COVID-19 morbidity and mortality.

## Introduction

Intensive postmarketing surveillance of Severe Acute Respiratory Syndrome Coronavirus 2 (SARS-CoV-2) vaccines is ongoing worldwide to provide updated information on their effectiveness and safety, thereby supporting regulatory benefit/risk assessment. Since early phase of the global vaccination campaign, case series [1–3] and pharmacovigilance reports [4,5] on myocarditis and pericarditis following the Coronavirus Disease 2019 (COVID-19) mRNA vaccine administration were published. Both events were included as related to COVID-19 disease in the early and updated Priority List of COVID-19 Adverse events of special interest, developed by Brighton Collaboration Group and Safety Platform for Emergency vACcines (SPEAC), in order to harmonize safety assessment of COVID-19 vaccines in pre- and postmarketing setting [6]. Moreover, as per core requirements for risk management plan (RMP), they have been periodically monitored through routine pharmacovigilance activities in the Monthly Summary Safety Reports of all COVID-19 vaccines [7].

On July 2021, the COVID-19 subcommittee of WHO Global Advisory Committee on Vaccine Safety reported that very rare cases of myocarditis and pericarditis had occurred more

often in adolescents or young adults and after the second dose, especially within a few days after COVID-19 mRNA vaccines, and encouraged countries to strengthen the monitoring of myocarditis/pericarditis [8]. At the same time, EMA's Pharmacovigilance Risk Assessment Committee (PRAC) began an assessment on signals of myocarditis and pericarditis with BNT162b2 and mRNA-1273 vaccine and concluded that both cardiac conditions can occur in very rare cases following vaccination with the COVID-19 mRNA vaccines. Thus, the Committee recommended to update the product information and the RMP for these vaccines, together with a direct healthcare professional communication to raise awareness among healthcare professionals [9].

In October 2021, further data were available from the Nordic population-based register study on myocarditis and pericarditis in northern Europe that prompted some public health organizations in the Nordic countries (e.g., Sweden, Finland, Norway, Iceland) [10] either to pause the use of the mRNA-1273 or to recommend the use of the BNT162b2 rather than mRNA-1273 in younger people and/or younger males. In December 2021, the PRAC reassessed the relevant safety signal, based on the Nordic study and on a study conducted using data from the French national health system (Epi-phare) [11], concluding that the risk for both events is overall "very rare" (up to 1 in 10,000 vaccinated people) and greater in younger males. A further update of product information was recommended, while the benefit/risk was confirmed as positive for the whole indications [12].

In line with these findings, recent published data from large population-based studies from Israel, United States, United Kingdom, and Denmark documented that the risks of myocarditis/pericarditis following mRNA vaccines differ by age groups, sex, and vaccine product with a higher risk in those younger than 40 years (S1 Table) [13–18].

In Italy, SARS-CoV-2 vaccines have been administered since late December 2020 and have been offered to the population according to a priority scheme, considering profession, age, and health conditions. Vaccination in adolescent (≥12 years) started on 31 May and 28 July 2021 for BNT162b2 and mRNA-1273, respectively.

Along with the enhanced passive surveillance of the Italian PharmacoVigilance network, an active surveillance, based on regional healthcare claims databases, was set up by the Italian National Institute of Health (ISS) and the Italian Medicines Agency (AIFA) to provide real-world data on SARS-CoV-2 vaccine safety.

To our knowledge, studies examining the association between mRNA-based COVID-19 vaccines in the population resident in Italy have not been published yet. Previous published studies have been conducted in other countries [13–18] and few of them have estimated risks in younger than 40 years by sex and age [13,14,17], while none of them have used a SCCS study design, with the exception of the study of Patone and colleagues [17]. The present study, while attempting to address these gaps, has the objective to investigate the association between mRNA-based COVID-19 vaccines (BNT162b2 and mRNA-1273) and myocarditis/pericarditis in the population of vaccinated persons aged 12 to 39 years, by age and sex, in Italy, during the period 27 December 2020 and 30 September 2021.

## Methods

### Data source

The active surveillance is based on a dynamic multiregional observational cohort. A distributed analysis framework is applied using *TheShinISS*, an R-based open-source statistical tool, developed by the National Institute of Health [19], that locally processes data collected and updated periodically from regional healthcare databases according to an ad hoc, study-tailored, common data model.

Data on vaccination exposure, on hospitalization for myocarditis/pericarditis, and on participant characteristics were retrieved from several routinely collected regional healthcare claims databases:

- COVID-19 vaccination registry to identify information on administered vaccines (product, date of administration, and doses for all vaccinated participants);

- population registry to identify information on age, sex, and vital status (causes of death are not recorded in this registry);

- hospital discharge and emergency care visit databases to identify myocarditis/pericarditis events (pre and post vaccination) and information on the comorbidities of the study participants in the period preceding the vaccination;

- pharmacy claims and copayment exemptions databases to obtain information on the comorbidities of the study participants in the period preceding the vaccination;

- vaccination registry to identify other vaccinations (e.g., flu and pneumococcal vaccines) administered in the period pre- and post-COVID-19 vaccination;

- COVID-19 surveillance system to obtain information on SARS-Cov-2 infection and related outcomes.

## Study design

We used a self-controlled case series (SCCS) design [20–24]. The SCCS design has emerged as a key methodology for studying the safety of vaccines and medicines. This approach only requires information from individuals who have experienced the event of interest and automatically controls for multiplicative time-invariant confounders, even when these are unmeasured or unknown. Originally designed to analyze the association between vaccination and specific events under the key assumption that events do not influence post-event exposures, this method has been adapted to event-dependent exposures, for example, when occurrence of an event may preclude any subsequent exposure (SCCS method for censored, perturbed, or curtailed post-event exposures) [23–25]. This is the case in observational studies of vaccines when the event of interest could be a contraindication to vaccination.

By using the adapted SCCS method for event-dependent exposures, we estimated the relative incidence (RI) of myocarditis/pericarditis following prespecified windows at risk after vaccination, in a within-person comparison of different time periods. The method allows for the control of all time-independent characteristics of participants. The SCCS method allows also for adjustment of potential time-varying confounders such as seasonal variation in risks.

## Study period and population

We investigated the association between mRNA-based COVID-19 vaccines and subsequent onset of myocarditis/pericarditis in the population aged 12 to 39 years in the period 27 December 2020 to 30 September 2021 (the latest date for which outcome data were available). Regional claims data were locally transformed into a study-specific common data model and locally processed using *TheShinISS*.

In the end, regional pseudonymized datasets were provided to the National Institute of Health for centralized analysis, in compliance with EU General Data Protection Regulation. Over the last 2 years, *TheShinISS* framework has been employed in several large scale observational studies exploring the association between several exposures and COVID-19 onset/

prognosis as well as other drug and vaccine-related research topics and is currently maintained by a collaborative research network [19,26–28]. The relational scheme of the study databases as well as *TheShinISS* flow diagram is described in S1 Fig.

Four Italian Regions (northern Italy: Lombardia, Veneto, Friuli Venezia Giulia; central Italy: Lazio), representing 36% of the population aged 12 to 39 years resident in Italy, contributed data of all vaccinated persons in this age group, in a period ranging from 27 December 2020 to the latest date for which data on outcomes were available, which varied across Regions: Lombardia up to 30 September 2021, Veneto up to 20 June 2021, Friuli Venezia Giulia up to 31 August 2021, and Lazio up to 16 June 2021). We included in the study all the persons aged 12 to 39 years who received a first dose of mRNA vaccines and were admitted to emergency care or hospital with the outcomes of interest. We excluded individuals with missing or inconsistent information on relevant variables (age, sex, vaccine product and dose, date of vaccination, of death and of event). Furthermore, we excluded individuals with a history of myocarditis or pericarditis within 365 days leading up to the start of the study period. The observation period for each case ranged from 27 December 2020 to the end of follow-up, which occurred at the time of death or at the end of Region-specific study period, whichever came first.

### Definition of outcomes

The outcome of interest was the first diagnosis of myocarditis/pericarditis identified through emergency care and/or hospital admission occurring between 27 December 2020 and 30 September 2021 using International Classification of Disease, Ninth Revision, Clinical Modification (ICD-9-CM codes of myocarditis: 391.2; 398.0; 422; 429.0; ICD-9-CM codes of pericarditis: 391.0; 393; 420; 423.1; 423.2; 423.9).

### Definition of exposures

The exposures of interest were the first or second dose of BNT162b2 and mRNA-1273 vaccines. The exposure risk period was not prespecified in the protocol submitted to the Ethical Committee and it was decided before data collection and analysis. It was defined as [0–21) days after first or second dose administration (vaccination date), which included day 0, the day of vaccination. The risk period was further subdivided into prespecified time periods: [0 to 7), [7 to 14), and [14 to 21) after each exposure date. The unexposed baseline period (reference period) was defined as any time of observation out of the risk periods (S2 Fig).

### Statistical analysis

Characteristics of the cohort of vaccinated persons and myocarditis and pericarditis cases were described by age, sex, and comorbidities. Temporal timing of myocarditis or pericarditis events in relation to first/second dose vaccination dates was described by week.

The SCCS model was fitted using unbiased estimating equations to estimate the RIs and their 95% confidence intervals (95% CI). In the following, we will use the term "association" between vaccine exposure and the study event (overall and in a given subgroup) for an RI estimate whose CI does not include the null effect. To handle event-dependent exposures, the SCCS model was properly modified considering a counterfactual exposure history for any exposures arising after occurrence of an event [20,24]. Five 45-day calendar periods were considered as time-varying covariate controlling for the seasonal effect (adjusted model). We also estimated the excess of cases (EC) per 100,000 vaccinated with 95% CIs applying nonparametric bootstrapping (10,000 replications) [29]. We carried out subgroup analyses by age group (12 to 17, 18 to 29, 30 to 39 years), sex, and vaccine product (BNT162b2 and mRNA-1273). To

assess the robustness of the primary analysis, the following sensitivity analyses regarding the modified SCCS method were conducted: (a) observation/exposure time period—we restricted the analysis to the study period from 27 December 2020 to 31 May 2021 [20,30] and to the study period from 1 June 2021 to 30 September 2021; we repeated the primary analysis excluding day 0 from the [0 to 7] day risk interval; we extended the exposure period to 28 days as well reducing it to 14 days; (b) heterologous vaccination—we carried out the primary analysis excluding individuals who received 2 different vaccine products at the first and second dose or censoring at time of second dose individuals who received a different vaccine product at the second dose (in the primary analysis, the second dose was assumed to be of the same product as the first one); (c) SARS-CoV-2 infection—we restricted the analyses to participants without a SARS-CoV-2-positive test before the occurrence of the event (any time) and within 10 days after the event. Further sensitivity analyses were performed exploring different assumptions of the standard SCCS method; (d) beginning observation at exposure; (e) beginning observation at time 0; (f) with prerisk period; (g) removing post event exposure. We conducted also an ancillary analysis reproducing the primary SCCS analysis in the vaccinated persons aged over 40 years.

The analyses were performed using R version 4.1.2 (R Core Team 2021) with SCCS package [31] and STATA version 16.1. This study is reported as per the REporting of studies Conducted using Observational Routinely-collected Data for Pharmacoepidemiology (RECORD-PE) checklist (S1 Checklist).

## Ethics and permissions

This study was approved by the National Unique Ethics Committee for the evaluation of clinical trials of medicines for human use and medical devices for patients with COVID-19 of the National Institute for Infectious Diseases "Lazzaro Spallanzani" in Rome (ordinance n. 335, 17/05/2021 and n. 399, 02/09/2021).

## Results

Our cohort included 13,728,174 persons older than 12 years, who received COVID-19 vaccines between 27 December 2020 to 30 September 2021, of these 10,769,025 (78.4%) received mRNA vaccines.

During the study period, 5,109,231 doses of mRNA vaccines were administered to 2,861,809 persons aged 12 to 39 years (median age 26 years, interquartile range, IQR 19 to 33; 49% females); 2,405,759 (84%) persons received BNT162b2 vaccine and 456,050 (16%) received mRNA-1273 vaccine. Among 24,469,038 doses uploaded on *TheShinISS*, the proportion of missing or inconsistent observations was 0.7% (n. 172,174) (S3 Fig). The vaccinated persons had a median follow-up time of 120 days (IQR 52 to 185). Characteristics of mRNA-vaccinated population aged 12 to 39 years and definition of study comorbidities are reported in S2 and S3 Tables, respectively.

During the study period, 441 persons had an emergency care and/or hospital admission related to myocarditis/pericarditis. Of these, 302 (68.5%) were males and 139 (31.5%) were females; there were 346 (78.5%) cases in those vaccinated with BNT162b2 and 95 (21.5%) in those vaccinated with mRNA-1273 (Table 1). Fig 1 describes the temporal trend of the occurrence of the events relative to vaccination date. We observed 1 death, for unknown cause, after 38 days following a pericarditis case that occurred 53 days after the second dose of BNT162b2 vaccine (unexposed period). The median follow-up time after the event was 93 days (IQR 56 to 113).

**Table 1. Characteristics of cases of myocarditis/pericarditis (n. 441) among the mRNA-vaccinated population aged 12–39 years from 27 December 2020 to 30 September 2021 by vaccine product.**

| | mRNA [n.(%)] | BNT162b2 [n.(%)] | mRNA-1273 [n.(%)] |
|---|---|---|---|
| **Number of participants** | **441** | **346** | **95** |
| **Sex** | | | |
| Males | 302 (68.5%) | 232 (67.1%) | 70 (73.7%) |
| Females | 139 (31.5%) | 114 (32.9%) | 25 (26.3%) |
| **Charlson index** | | | |
| ≥1 | 53 (12.0%) | 46 (13.3%) | 7 (7.4%) |
| **Hospitalizations in the last 2 years** | | | |
| ≥1 | 188 (42.6%) | 154 (44.5%) | 34 (35.8%) |
| **Comorbidities** | | | |
| COVID-19 diagnosis before vaccination | 60 (13.6%) | 48 (13.9%) | 12 (12.6%) |
| COPD/Asthma | 32 (7.3%) | 28 (8.1%) | 4 (4.2%) |
| Chronic pulmonary disease | 16 (3.6%) | 15 (4.3%) | 1 (1.1%) |
| Chronic kidney failure | 9 (2.0%) | 7 (2.0%) | 2 (2.1%) |
| Neoplasms | 16 (3.6%) | 14 (4.0%) | 2 (2.1%) |
| Diabetes mellitus | 8 (1.8%) | 4 (1.2%) | 4 (4.2%) |
| Hematologic disease | 50 (11.3%) | 43 (12.4%) | 7 (7.4%) |
| Cardiovascular/cerebrovascular diseases | 170 (38.5%) | 138 (39.9%) | 32 (33.7%) |
| Hypertension | 82 (18.6%) | 66 (19.1%) | 16 (16.8%) |
| Hepatopathy | 2 (0.5%) | 1 (0.3%) | 1 (1.1%) |
| HIV | 2 (0.5%) | 2 (0.6%) | 0 |
| Rheumatic diseases | 23 (5.2%) | 18 (5.2%) | 5 (5.3%) |
| Cystic fibrosis | 0 | 0 | 0 |
| Neurological diseases | 26 (5.9%) | 20 (5.8%) | 6 (6.3%) |
| Peptic ulcer | 152 (34.5%) | 122 (35.3%) | 30 (31.6%) |
| Colitis | 5 (1.1%) | 4 (1.2%) | 1 (1.1%) |
| Celiac disease | 1 (0.2%) | 1 (0.3%) | 0 |
| Infection | 155 (35.1%) | 125 (36.1%) | 30 (31.6%) |
| **Prior drugs use** | | | |
| Prescriptions in the last 12 months (1+) | 307 (69.6%) | 246 (71.1%) | 61 (64.2%) |
| Corticosteroids for systemic use | 57 (12.9%) | 49 (14.2%) | 8 (8.4%) |
| NSAIDs use | 48 (10.9%) | 40 (11.6%) | 8 (8.4%) |
| Estroprogestinics use | 2 (0.5%) | 1 (0.3%) | 1 (1.1%) |

COPD, chronic obstructive pulmonary disease; HIV, human immunodeficiency virus; n., number; NSAID, nonsteroidal anti-inflammatory drug; yrs, years.

Table 2 reports the results of the primary analysis from the SCCS model, with RIs adjusted by calendar period, for the 441 cases aged 12 to 39 years. The unadjusted estimates of RI are shown in S4 Table. ECs are reported for RIs with 95% CI not including the null effect. During the 21-day risk interval, there were a total of 114 cases of myocarditis/pericarditis (74 with BNT162b2 and 40 with mRNA-1273), corresponding to RIs of 1.08 (95% CI: 0.70 to 1.67) and 1.99 (95% CI: 1.30 to 3.05) after first and second dose of BNT162b2, respectively, and 2.22 (95% CI 1.00 to 4.91) and 2.63 (95% CI 1.21 to 5.71) after first and second dose of mRNA-1273, respectively. The majority of these cases occurred within the [0 to 7) day risk period after the first or second dose administration of mRNA vaccines (n. 70, 61.4%). An increased risk of myocarditis/pericarditis [0 to 7) days following a first dose of mRNA-1273 was observed (RI = 6.55, 95% CI: 2.73 to 15.72), while no association was found with BNT162b2. An

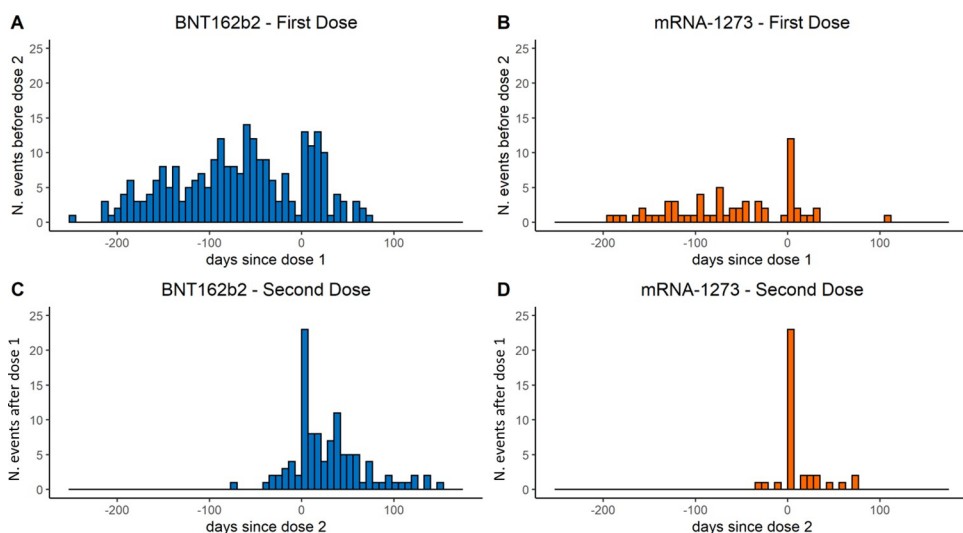

**Fig 1. Days from mRNA vaccination to myocarditis/pericarditis in population aged 12–39 years by vaccine product and dose.** Top (panel A-BNT162b and B-mRNA-1273): days from dose 1, for events occurring before dose 2 if present, or at any time if dose 2 not present. Bottom (panel C-BNT162b and D-mRNA-1273): days from dose 2, for events occurring after dose 1. *Each bar corresponds to 1 week starting from day 0.

increased risk of myocarditis/pericarditis [0 to 7) days was also observed following a second dose of BNT162b2 (RI = 3.39, 95% CI: 2.02 to 5.68) and mRNA-1273 (RI = 7.59, 95% CI: 3.26 to 17.65). Over the [0 to 7) days postvaccination, we estimated an additional 2.0 (95% CI: 0.8 to 3.6) myocarditis/pericarditis cases per 100,000 vaccinated persons following the first dose of mRNA-1273; following a second dose of the BNT162b2 and mRNA-1273, over the [0 to 7) days post vaccination, we estimated an additional 0.8 (95% CI: 0.4 to 1.4) and 5.5 (95% CI: 3.0 to 7.9) myocarditis/pericarditis cases per 100,000 vaccinated, respectively.

**Table 2. Adjusted RI estimated by SCCS and excess cases per 100,000 vaccinated by vaccine product and risk intervals: 346 myocarditis/pericarditis events in the BNT162b2 and 95 events in the mRNA-1273 vaccinated population aged 12–39 years from 27 December 2020 to 30 September 2021.**

| Risk interval | Dose | BNT162b2 (n. 346) | | | mRNA-1273 (n. 95) | | |
|---|---|---|---|---|---|---|---|
| | | Events in the risk interval (n) | Adjusted RI (95% CI)* | Excess cases per 100,000 vaccinated (95% CI)** | Events in the risk interval (n) | Adjusted RI (95% CI)* | Excess cases per 100,000 vaccinated (95% CI)** |
| [0–21) | Dose 1 | 35 | 1.08 (0.70–1.67) | | 15 | 2.22 (1.00–4.91) | 1.8 (−0.2–3.7) |
| | Dose 2 | 39 | 1.99 (1.30–3.05) | 1.0 (0.3–1.7) | 25 | 2.63 (1.21–5.71) | 4.2 (0.8–7.2) |
| [0–7) | Dose 1 | 14 | 1.27 (0.70–2.31) | | 11 | 6.55 (2.73–15.72) | 2.0 (0.8–3.6) |
| | Dose 2 | 22 | 3.39 (2.02–5.68) | 0.8 (0.4–1.4) | 23 | 7.59 (3.26–17.65) | 5.5 (3.0–7.9) |
| [7–14) | Dose 1 | 10 | 0.92 (0.46–1.82) | | 3 | 1.58 (0.45–5.58) | |
| | Dose 2 | 7 | 1.07 (0.50–2.30) | | 0 | — | |
| [14–21) | Dose 1 | 11 | 1.09 (0.56–2.12) | | 1 | 0.49 (0.06–4.07) | |
| | Dose 2 | 10 | 1.58 (0.78–3.21) | | 2 | 0.71 (0.17–3.09) | |
| Ref. | | 272 | 1.0 | | 55 | 1.0 | |

*Adjusted by calendar period.

**Excess cases are not given when the 95% CI of RI included the null effect.

CI, confidence interval; n., number; Ref., reference period (unexposed period); RI, relative incidence; SCCS, self-controlled cases series.

**Table 3. Adjusted RI and excess of cases per 100,000 vaccinated in the 7-day risk periods after mRNA vaccination in the vaccinated population aged 12–39 years from 27 December 2020 to 30 September 2021 by sex, age group, and vaccine product.**

| Age, years | Sex | Risk interval | Dose | BNT162b2 (n. 346) | | | mRNA-1273 (n. 95) | | |
|---|---|---|---|---|---|---|---|---|---|
| | | | | Events in the risk interval (n) | Adjusted RI (95% CI)* | Excess cases per 100,000 vaccinated (95% CI)** | Events in the risk interval (n) | Adjusted RI (95% CI)* | Excess cases per 100,000 vaccinated (95% CI)** |
| 12–39 | Males +Females | [0–7] | Dose 1 | 14 | 1.27 (0.70–2.31) | | 11 | 6.55 (2.73–15.72) | 2.0 (0.8–3.6) |
| | | | Dose 2 | 22 | 3.39 (2.02–5.68) | 0.8 (0.4–1.4) | 23 | 7.59 (3.26–17.65) | 5.5 (3.0–7.9) |
| | | | Ref. | 272 | 1 | | 55 | 1 | |
| | Males | [0–7] | Dose 1 | 9 | 1.53 (0.71–3.31) | | 10 | 12.28 (4.09–36.83) | 3.8 (1.5–6.3) |
| | | | Dose 2 | 13 | 3.45 (1.78–6.68) | 1.0 (0.3–1.8) | 19 | 11.91 (3.88–36.53) | 8.8 (4.9–12.9) |
| | | | Ref. | 184 | 1 | | 38 | 1 | |
| | Females | [0–7] | Dose 1 | 5 | 0.88 (0.34–2.32) | | 1 | 0.69 (0.08–5.75)*** | |
| | | | Dose 2 | 9 | 3.38 (1.47–7.74) | 0.7 (0.1–1.4) | 4 | 2.08 (0.45–9.72)*** | |
| | | | Ref. | 88 | 1 | | 17 | 1 | |
| 12–17 | Males +Females | [0–7] | Dose 1 | 3 | 1.06 (0.17–6.59) | | 0 | **** | |
| | | | Dose 2 | 7 | 5.74 (1.52–21.72) | 1.7 (0.04–3.2) | 3 | **** | |
| | | | Ref. | 31 | 1 | | 7 | 1 | |
| 18–29 | Males +Females | [0–7] | Dose 1 | 7 | 1.76 (0.76–4.05) | | 9 | 7.58 (2.62–21.94) | 3.4 (1.1–6.0) |
| | | | Dose 2 | 11 | 4.02 (1.81–8.91) | 1.1 (0.2–2.0) | 18 | 9.58 (3.32–27.58) | 8.6 (4.4–12.6) |
| | | | Ref. | 121 | 1 | | 28 | 1 | |
| 30–39 | Males +Females | [0–7] | Dose 1 | 4 | 0.86 (0.31–2.38) | | 2 | 6.57 (1.32–32.63) | 1.0 (^NE–3.3) |
| | | | Dose 2 | 4 | 1.64 (0.59–4.53) | | 2 | 3.22 (0.69–15.10) | |
| | | [7–14] | Dose 1 | 4 | 0.97 (0.34–2.80) | | 2 | 5.87 (1.34–25.74) | 1.0 (^NE–3.3) |
| | | | Ref. | 120 | 1 | | 20 | 1 | |

*Adjusted by calendar period.

**Excess cases are not given when the 95% CI of RI included the null effect.

***Unadjusted RIs due to the small number of cases.

****Considering the small number of cases, it was not possible to provide any estimates.

^Considering the small number of cases, bootstrapping produced implausible results for inferior limit of 95% CIs.

CI, confidence interval; NE, not estimable; n., number; Ref., reference period (unexposed baseline period); RI, relative incidence.

## Subgroup analysis by sex and age group

Table 3 and Fig 2 show the adjusted RIs (unadjusted estimates in S5 Table) and ECs by age, sex, and product in the 7 days risk period (S6–S15 Tables and S4 Fig).

In males, the risk of myocarditis/pericarditis increased in the [0 to 7] days following a first dose of mRNA-1273 (RI = 12.28, 95% CI: 4.09 to 36.83) and following a second dose of BNT162b2 (RI = 3.45, 95% CI: 1.78 to 6.68) and mRNA-1273 (RI = 11.91, 95% CI: 3.88 to 36.53). In females, we found an increased risk of myocarditis/pericarditis [0 to 7] days following a second dose of BNT162b2 (RI = 3.38, 95% CI: 1.47 to 7.74), while no association was observed with mRNA-1273.

In males, we estimated an additional 3.8 (95% CI: 1.5 to 6.3) EC per 100,000 in the [0 to 7] days following a first dose of mRNA-1273, and an additional 1.0 (95% CI: 0.3 to 1.8) and 8.8

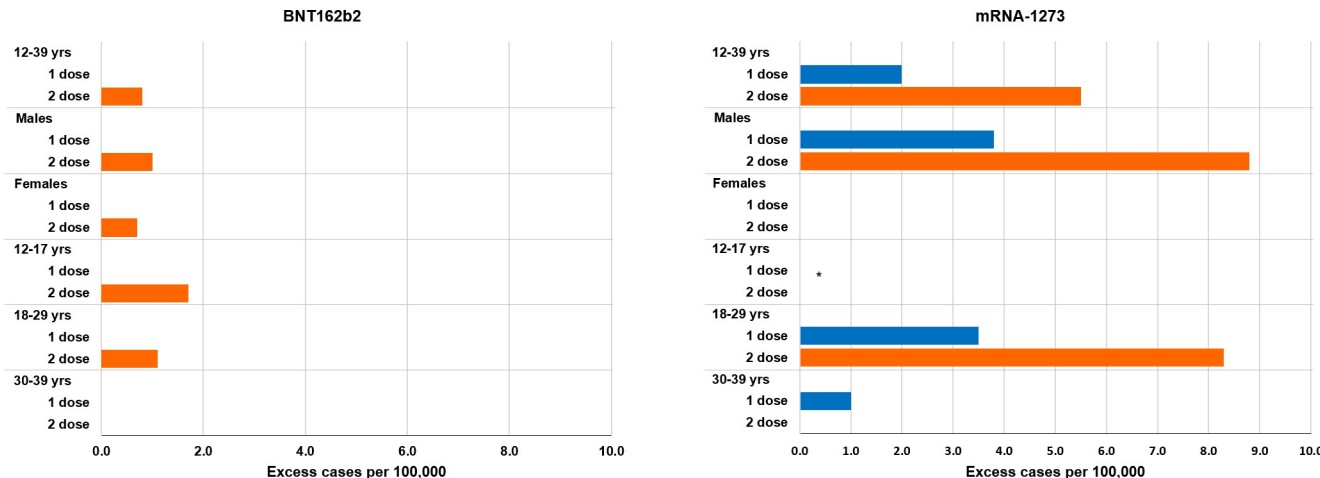

**Fig 2. Excess of cases per 100,000 vaccinated in the [0–7) days risk period following BNT162b2 and mRNA-1273 vaccination in the vaccinated population aged 12–39 years from 27 December 2020 to 30 September 2021, by sex, age group, and dose (first dose blue, second dose orange).**
*Considering the small number of cases in the vaccinated with mRNA-1273 of age 12–17 years, it was not possible to provide any estimates; excess cases are not given when the 95% CI of RI included the null effect over the [0–7) day risk interval post vaccination. RI, relative incidence; yrs, years.

(95% CI: 4.9 to 12.9) EC per 100,000 in the [0 to 7) days following a second dose of BNT162b2 and mRNA-1273, respectively. In females, we estimated an additional 0.7 (95% CI: 0.1 to 1.4) EC per 100,000 in the [0 to 7) days following a second dose of BNT162b2.

In the analyses by age group, we estimated an increased risk of myocarditis/pericarditis [0 to 7) days following the second dose of BNT162b2 (RI = 5.74, 95% CI: 1.52 to 21.72) in those aged 12 to 17 years. Number of events was insufficient to fit the SCCS model with mRNA-1273 aged 12 to 17 years. Of note, there were 3 events after the second dose in the [0 to 7) day interval compared to 7 events in the reference period. In the 18 to 29 year age group, we observed an increased risk of myocarditis/pericarditis [0 to 7) days following a first and second dose of mRNA-1273 (RI = 7.58, 95% CI: 2.62 to 21.94 and RI = 9.58, 95% CI: 3.32 to 27.58, respectively) and following the second dose of BNT162b2 (RI = 4.02, 95% CI: 1.81 to 8.91). In the age group 30 to 39 years, we found an increased risk of myocarditis/pericarditis [0 to 7) days (RI = 6.57, 95% CI: 1.32 to 32.63) and [7 to 14) days (RI = 5.87, 95% CI: 1.34 to 25.74) following the first doses of mRNA-1273, while no association was observed with BNT162b2.

In the age group 12 to 17 years, we estimated an additional 1.7 (95% CI: 0.04 to 3.2) EC per 100,000 in the [0 to 7) days following a second dose of BNT162b2. In the age group 18 to 29 years, we estimated an additional 3.4 (95% CI: 1.1 to 6.0) EC per 100,000 in the [0 to 7) days following a first dose of mRNA-1273; an additional 1.1 (95% CI: 0.2 to 2.0) and 8.6 (95% CI: 4.4 to 12.6) EC per 100,000 in the [0 to 7) days following a second dose of BNT162b2 and mRNA-1273, respectively. In the age group 30 to 39 years, we estimated an additional 1.0 EC per 100,000 (95% CI: not estimable-3.3) both in the [0 to 7) days and [7 to 14) days following the first dose of mRNA-1273, respectively.

## Sensitivity and ancillary analyses

All sensitivity analyses, using the modified SCCS method for event-dependent exposure, were consistent with the main results of the study (S16 Table). The sensitivity analysis that was conducted to highlight the potential effect of notoriety bias (by restricting the observation period before and after 31 May 2021) indicated that RIs estimates and CIs are largely overlapping, even though we cannot rule out a slight inflation of the estimates in the second period.

Additional sensitivity analyses, based on the standard SCCS model, showed an inflation of the estimates, with the exception of the first analysis (standard SCCS beginning observation at exposure) usually used as an alternative approach to the modified SCCS method (S17 Table).

The ancillary analysis on 2,050 cases aged over 40 years (BNT162b2 n. 1,759; mRNA-1273 n. 291) did not show an increase risk of myocarditis/pericarditis for BNT162b2 and mRNA-1273 after 7 days following the first dose (RI = 0.59, 95% CI: 0.42 to 0.82 and RI = 0.56, 95% CI: 0.23 to 1.36) and the second dose (RI = 0.84, 95% CI: 0.61 to 1.16 and RI = 1.11, 95% CI: 0.57 to 2.17) (S18 Table).

## Discussion

### Principal findings

This first Italian large population-based study covering about 3 million of vaccinated persons aged 12 to 39 years found an association between myocarditis/pericarditis within a week following each dose of mRNA vaccines.

The risk of myocarditis/pericarditis is particularly higher after 7 days following the first or second dose of mRNA-1273 vaccine in the overall population. Subgroup analysis by sex suggested that the increased risk was present only in males after both the first and second dose with 3.8 and 8.8 EC per 100,000 vaccinated, respectively. Stratifying by age, greater risks were found in those aged 18 to 29 years with EC of 3.4 and 8.6 per 100,000 following the first and the second doses, respectively. In the age group 12 to 17 years, the number of events was insufficient for risk estimate.

We also observed an association between BNT162b2 and myocarditis/pericarditis, but only in the 7 days following the second dose, where the risks remain similar between males and females with 1.0 and 0.7 EC per 100,000 vaccinated, respectively. In the age groups 12 to 17 years and 18 to 29 years, where the increased risks were confined, the estimated EC were 1.7 and 1.1 per 100,000 vaccinated, respectively.

Vaccine-associated acute myocarditis is generally attributable to allergic/hypersensitivity reactions as observed in other vaccines [32]. However, the pathophysiology of myocarditis and pericarditis associated to mRNA vaccines is not clearly understood and different mechanisms have been postulated. Molecular mimicry between the spike protein and self-antigens [33], trigger of preexisting dysregulated immune pathways, immune response to mRNA [34] or dysregulated cytokine expression [35] have been proposed.

Our results on the increased risk in the 7 days after each dose of mRNA-1273 and the second dose of BNT162b2 are consistent with the onset of viral myocarditis symptoms often reported in the first week from the infection [32,36].

Moreover, it has been postulated that a very high antibody response to mRNA vaccines in predisposed young people may elicit an uncontrolled inflammatory response similar to multisystem inflammatory syndrome observed in children (MIS-C) with SARS-CoV-2 infection [37]. To date, no clear evidence is available, and further studies are needed to clarify which is the exact mechanism of mRNA vaccines-related myocarditis and pericarditis.

Furthermore, our observation on the increased risk in young males resembles classical epidemiological features of myocarditis due to other causes [38], included COVID-19-related myocarditis [39], but the exact role of age and sex is still unclear. In a recent review, a possible effect of sex hormones in immune response is summarized, with a possible role of testosterone by a combined mechanism of inhibition of anti-inflammatory cells and commitment to a Th1-type immune response in male and of inhibitory effects of estrogen on proinflammatory T cells in female [40].

## Comparison with related studies

In line with a previous US study [15], we identified an association between mRNA vaccines and myocarditis/pericarditis in individuals younger than 40 years within the 0- to 7-day period following the first and the second dose.

Our results are also consistent with observational studies that documented markedly increased risk of myocarditis in England [16] and myocarditis or myopericarditis in Denmark [18] in the population vaccinated with mRNA-1273. Specifically, in the Danish study, it was reported a strong association between mRNA-1273 and myocarditis or myopericarditis within 28 days from vaccination (hazard ratio, HR = 5.24; 2.47 to 11.12) with an estimated 5.7 EC per 1,000,000 vaccinated. The UK study also suggested a strong association within the 1 to 28 days after first and second mRNA-1273 dose (incidence rate ratio, IRR = 3.89; 1.60 to 9.44; and 20.71, 4.02 to 106.68; respectively) corresponding to 8 and 15 EC per 1,000,000 vaccinated [16]. A recent updated SCCS analysis of English data (preprint publication) [17], stratified by age and sex, also reported a higher risk in male aged less than 40 years (first dose IRR = 2.34; 1.03 to 5.34; second dose IRR = 16.52; 9.10 to 30.00) corresponding to EC of 12 and 101 per 1,000,000, respectively; a markedly increased risk was also observed in females after the second dose of mRNA-1273 (IRR = 7.55; 1.67 to 34.12) with 8 EC per 1,000,000 vaccinated [17]. Our findings are in line with a higher risk observed with mRNA-1273 vaccine in a recently published study conducted in Denmark, Finland, Norway, and Sweden [41].

Results on the association between BNT162b2 and myocarditis/pericarditis are less conclusive. We found an association in the 7 days after the second dose both in males and females. Findings from Israel [14] and England [17] confirmed an association in adolescent and adult males younger than 40 years, but not in female participants. Particularly, the English study, including data on the third dose of BNT162b2, highlighted that in males 12 to 39 years, the risk sequentially increased following each dose of vaccine (IRR = 1.66, 3.41, and 7.60, respectively) with an EC of 3, 12, and 13 per 1,000,000 vaccinated, respectively. No association was found in females and in males older than 40 years [17].

Conversely, a population-based study conducted in Denmark [18], with a more stringent case definition, did not support the association between BNT162b2 and myocarditis or myopericarditis in the 28 days after vaccination, both overall and in the 12 to 39 year group, but an association only in females (HR = 3.73; 1.82 to 7.65) was found.

## Strengths and limitations

Our study strengths include the large sample size, the broad geographical distribution of the cohort and the availability of data on COVID-19 vaccination and outcomes, and comorbidities and patients' demographic characteristics from healthcare databases. The large sample size (about 3 million vaccinated people aged 12 to 39 years) allowed us to look at fine risk intervals following vaccinations and conduct several subgroup analyses. Since data were collected from routinely collected data in the claims databases, irrespective of research question, there is no potential for recall or selection bias.

An interesting methodological point of our study is the choice of the SCCS method modified to handle event-dependent exposures. In a very recent paper on the use of the SCCS method with application to COVID-19 vaccine safety [25], the authors quantify, by simulation, the overestimation of risk in the standard SCCS method when vaccination is severely delayed or canceled after the occurrence of an event. They argue that when vaccination is only delayed by a short time, this bias may be corrected within the standard SCCS methodology by including a prevaccination risk period. Instead, the modified SCCS model for event-dependent exposures needs to be applied when vaccination may be severely delayed or canceled entirely. In

addition, they discuss the usefulness of including unvaccinated cases in the analysis, this inclusion is deemed to be necessary to avoid once again overestimation of the risk estimates. This was not possible in our study since the surveillance collected data only on vaccinated cases. However, it is also shown in the paper that the presence of an appreciable proportion of vaccinated cases for whom the event occurs before the first vaccine dose, mitigated this effect. In their simulation, where the above proportion was about 50%, the overestimation of risk was only approximately 10%. In our study this proportion is 48%, and a slight inflation of the relative risk cannot be excluded. It is reassuring that the sensitivity analysis (S17 Table) in which the observation time began at first or second vaccination dose in a standard SCCS method gave similar estimates to our main analysis.

However, our study also has limitations. First, there is the possibility of notoriety bias due to overdiagnosis of cases of myocarditis/pericarditis because of the increased public and medical awareness of these potential adverse events following mRNA vaccination. The effect observed in the sensitivity analysis could be partly explained by a different age profile and characteristics of the 2 vaccinated population before and after 31 May 2021. Second, diagnoses of myocarditis and pericarditis were retrieved from hospital discharge and emergency care visit databases, and they were not validated through the review of clinical records. For this reason, a misclassification of the outcomes that biased the association cannot be excluded. Third, we did not collect further information to assess the severity of the outcomes. To date, data were collected without differentiating between emergency care admission and hospital admission, and length of the hospitalization was not available. Lastly, although the large sample size including about 3 million of vaccinated persons, due to the small number of events, it was not possible to provide robust model estimates in some subgroup analyses (i.e., mRNA-1273 in the subgroup of adolescents aged 12 to 17 years).

In conclusion, this population-based study suggests that mRNA vaccines were associated with myocarditis/pericarditis in the population younger than 40 years. According to our results, the risk increased after the second dose of BNT162b2 vaccine and after both doses of mRNA-1273 vaccine. The highest risks were observed in males of 12 to 39 years and in males and females of 18 to 29 years of age vaccinated with mRNA-1273 vaccine. However, vaccine-associated risks should always be evaluated in the light of the proven vaccine effectiveness in preventing serious COVID-19 disease and death. After the evaluation of all data available, the Italian Medicines Agency (AIFA) continued to consider a positive benefit-risk profile of mRNA COVID-19 vaccines in this population.

Further monitoring of data from this active surveillance is needed to evaluate the relationship between mRNA vaccines and myocarditis/pericarditis by age within sex, including population of children (5 to 11 years old) and the effect of the third dose (booster dose).

## Supporting information

**S1 Checklist. RECORD-PE Checklist.**
(PDF)

**S1 Table. Observational studies on safety of COVID-19 mRNA vaccines and myocarditis and/or pericarditis outcomes.**
(DOCX)

**S2 Table. Characteristics of mRNA-vaccinated population aged 12–39 years (n. 2,861,809) from 27 December 2020 to 30 September 2021, by vaccine product.** n., number; yrs, years; COPD, chronic obstructive pulmonary disease; HIV, human immunodeficiency virus;

NSAIDs, nonsteroidal anti-inflammatory drugs.
(DOCX)

**S3 Table. Definition of study comorbidities.** ATC, Anatomical Therapeutic Chemical Classification System; COPD, chronic obstructive pulmonary disease; HIV, human immunodeficiency virus; ICD, International Classification of Disease; NSAID, nonsteroidal anti-inflammatory drugs.
(DOCX)

**S4 Table. RI estimated by SCCS by vaccine product and risk intervals: 346 myocarditis/ pericarditis events in the BNT162b2 and 95 events in the mRNA-1273 vaccinated population aged 12–39 years from 27 December 2020 to 30 September 2021.** CI, confidence interval; n., number; Ref., reference period (unexposed period); RI, relative incidence; SCCS, self-controlled cases series.
(DOCX)

**S5 Table. RI estimated by SCCS in the [0–7) risk period after mRNA vaccination in the vaccinated population aged 12–39 years from 27 December 2020 to 30 September 2021 by sex, age group, and vaccine product.** *Considering the small number of cases, it was not possible to provide any estimates. CI, confidence interval; n., number; Ref., reference period (unexposed baseline period); RI, relative incidence.
(DOCX)

**S6 Table. Adjusted RI estimated by SCCS and excess cases per 100,000 vaccinated by risk intervals: 232 myocarditis and/or pericarditis events in the BNT162b2 vaccinated males aged 12–39 years from 27 December 2020 to 30 September 2021.** *Adjusted by calendar period. **Excess cases are not given when the 95% CI of RI included the null effect. CI, confidence interval; n., number; Ref., reference period (unexposed period); RI, relative incidence; SCCS, self-controlled cases series.
(DOCX)

**S7 Table. Adjusted RI estimated by SCCS and excess cases per 100,000 vaccinated by risk intervals: 114 myocarditis and/or pericarditis events in the BNT162b2 vaccinated females aged 12–39 years from 27 December 2020 to 30 September 2021.** *Adjusted by calendar period. **Excess cases are not given when the 95% CI of RI included the null effect. CI, confidence interval; n., number; Ref., reference period (unexposed period); RI, relative incidence; SCCS, self-controlled cases series.
(DOCX)

**S8 Table. Adjusted RI estimated by SCCS and excess cases per 100,000 vaccinated by risk intervals: 46 myocarditis and/or pericarditis events in the BNT162b2 vaccinated population aged 12–17 years from 27 December 2020 to 30 September 2021.** *Adjusted by calendar period. **Excess cases are not given when the 95% CI of RI included the null effect. CI, confidence interval; n., number; Ref., reference period (unexposed period); RI, relative incidence; SCCS, self-controlled cases series.
(DOCX)

**S9 Table. Adjusted RI estimated by SCCS and excess cases per 100,000 vaccinated by risk intervals: 154 myocarditis and/or pericarditis events in the BNT162b2 vaccinated population aged 18–29 years from 27 December 2020 to 30 September 2021.** *Adjusted by calendar period. **Excess cases are not given when the 95% CI of RI included the null effect. CI, confidence interval; n., number; Ref., reference period (unexposed period); RI, relative incidence;

SCCS, self-controlled cases series.
(DOCX)

**S10 Table. Adjusted RI estimated by SCCS and excess cases per 100,000 vaccinated by risk intervals: 146 myocarditis and/or pericarditis events in the BNT162b2 vaccinated population aged 30–39 years from 27 December 2020 to 30 September 2021.** *Adjusted by calendar period. **Excess cases are not given when the 95% CI of RI included the null effect. CI, confidence interval; n., number; Ref., reference period (unexposed period); RI, relative incidence; SCCS, self-controlled cases series.
(DOCX)

**S11 Table. Adjusted RI estimated by SCCS and excess cases per 100,000 vaccinated by risk intervals: 70 myocarditis and/or pericarditis events in the mRNA-1273 vaccinated males aged 12–39 years from 27 December 2020 to 30 September 2021.** *Adjusted by calendar period. **Excess cases are not given when the 95% CI of RI included the null effect. CI, confidence interval; n., number; Ref., reference period (unexposed period); RI, relative incidence; SCCS, self-controlled cases series.
(DOCX)

**S12 Table. Adjusted RI estimated by SCCS and excess cases per 100,000 vaccinated by risk intervals: 25 myocarditis and/or pericarditis events in the mRNA-1273 vaccinated females aged 12–39 years from 27 December 2020 to 30 September 2021.** *Adjusted by calendar period; **excess cases are not given when the 95% CI of RI included the null effect. CI, confidence interval; n., number; Ref., reference period (unexposed period); RI, relative incidence; SCCS, self-controlled cases series.
(DOCX)

**S13 Table. Adjusted RI estimated by SCCS and excess cases per 100,000 vaccinated by risk intervals: 11 myocarditis and/or pericarditis events in the mRNA-1273 vaccinated population aged 12–17 years from 27 December 2020 to 30 September 2021.** *Adjusted by calendar period. **Excess cases are not given when the 95% CI of RI included the null effect. §Considering the small number of cases in this age group, it was not possible to provide any estimates. CI, confidence interval; n., number; Ref., reference period (unexposed period); RI, relative incidence; SCCS, self-controlled cases series.
(DOCX)

**S14 Table. Adjusted RI estimated by SCCS and excess cases per 100,000 vaccinated by risk intervals: 57 myocarditis and/or pericarditis events in the mRNA-1273 vaccinated population aged 18–29 years from 27 December 2020 to 30 September 2021.** *Adjusted by calendar period. **Excess cases are not given when the 95% CI of RI included the null effect. CI, confidence interval; n., number; Ref., reference period (unexposed period); RI, relative incidence; SCCS, self-controlled cases series.
(DOCX)

**S15 Table. Adjusted RI estimated by SCCS and excess cases per 100,000 vaccinated by risk intervals: 27 myocarditis and/or pericarditis events in the mRNA-1273 vaccinated population aged 30–39 years from 27 December 2020 to 30 September 2021.** *Adjusted by calendar period. **Excess cases are not given when the 95% CI of RI included the null effect. CI, confidence interval; n., number; Ref., reference period (unexposed period); RI, relative incidence; SCCS, self-controlled cases series.
(DOCX)

**S16 Table. Sensitivity analyses.** *Adjusted by calendar period. **Only participants vaccinated and with event in each period were included in this analysis. ***Considering the small number of cases, it was not possible to provide any estimates. CI, confidence interval; SCCS, self-controlled cases series.
(DOCX)

**S17 Table. Sensitivity analyses: standard SCCS method.** *Adjusted by calendar period. CI, confidence interval; SCCS, self-controlled cases series.
(DOCX)

**S18 Table. Adjusted RI estimated by SCCS and excess cases per 100,000 vaccinated by risk intervals: 1,759 myocarditis and/or pericarditis events in the BNT162b2 and 291 events in the mRNA-1273 vaccinated population aged ≥40 years from 27 December 2020 to 30 September 2021 (ancillary analysis).** *Adjusted by calendar period. **Excess cases are not given when the 95% CI of RI included the null effect. CI, confidence interval; n., number; Ref., reference period (unexposed period); RI, relative incidence; SCCS, self-controlled cases series.
(DOCX)

**S1 Fig. Diagram showing the data flow when using *TheShinISS* to locally process healthcare data structured according to a CMD.** CDM, common data model.
(DOCX)

**S2 Fig. Schematic presentation of the SCCS method.** SCCS, self-controlled cases series.
(DOCX)

**S3 Fig. Flow chart of study population.**
(DOCX)

**S4 Fig. Adjusted RI in the [0–7) risk period after mRNA vaccination in the vaccinated population aged 12–39 years from 27 December 2020 to 30 September 2021 by vaccine product, sex, and age group.** (*) Considering the small number of cases in the vaccinated with mRNA-1273 of age 12–17 years, it was not possible to provide any estimates. CI, confidence interval; F, females; M, males; RI, relative incidence.
(DOCX)

## Acknowledgments

We would like to thank Gianpaolo Scalia Tomba, Giuseppe Traversa, Maria Paola Trotta, Maria Grazia Evandri, Nicola Magrini, and Patrizia Popoli for their useful suggestions and *TheShinISS-Vax|COVID Surveillance Group*: Francesca Menniti Ippolito, Roberto Da Cas, Ilaria Ippoliti, Marco Massari, Cristina Morciano, Paola Ruggeri, Emanuela Salvi, Stefania Spila Alegiani (National Centre for Drug Research and Evaluation, National Institute of Health—Istituto Superiore di Sanità); Anna Rosa Marra, Patrizia Felicetti, Pasquale Marchione, Fiorella Petronzelli, Giuseppe Pimpinella, Loriana Tartaglia (Department of postmarketing surveillance, Italian Medicines Agency—Agenzia Italiana del Farmaco); Patrizio Pezzotti, Antonino Bella, Massimo Fabiani, Matteo Spuri, Alberto Mateo Urdiales (Infectious Disease Department, National Institute of Health—Istituto Superiore di Sanità); Lorenza Ferrara, Luca Bolognesi, Lucia Favella (Piemonte Region); Giuseppe Monaco, Olivia Leoni, Michele Ercolanoni, Marco Lazzeretti (Lombardia Region); Gianluca Trifirò, Ugo Moretti, Giovanna Scroccaro, Paola Deambrosis, Giovanna Zanoni, Manuel Zorzi, Emanuela Bovo, Michele Tonon, Elena Vecchiato (Veneto Region); Paola Rossi, Cristina Zappetti, Sara Samez, Elena Clagnan (Friuli Venezia Giulia Region); Ester Sapigni, Aurora Puccini, Nazanin Morgheiseh (Emilia

Romagna Region); Marco Tuccori, Rosa Gini, Giulia Hyeraci, Valentina Borsi (Toscana Region); Lorella Lombardozzi, Valeria Desiderio, Nadia Mores, Valeria Belleudi, Maria Balducci, Francesca Romana Poggi (Lazio Region); Annalisa Capuano, Ugo Trama, Massimo Di Gennaro, Roberta Giordana, Maria Grazia Fumo (Campania Region); and Silvio Tafuri, Pasquale Stefanizzi, Domenica Ancona (Puglia Region).

## Author Contributions

**Conceptualization:** Marco Massari, Stefania Spila Alegiani, Cristina Morciano, Matteo Spuri, Pasquale Marchione, Patrizia Felicetti, Gianluca Trifirò, Roberto Da Cas, Fiorella Petronzelli, Anna Rosa Marra, Francesca Menniti Ippolito.

**Data curation:** Marco Massari, Stefania Spila Alegiani, Valeria Belleudi, Francesca Romana Poggi, Marco Lazzeretti, Michele Ercolanoni, Elena Clagnan, Emanuela Bovo, Roberto Da Cas.

**Formal analysis:** Marco Massari, Stefania Spila Alegiani, Cristina Morciano, Matteo Spuri, Valeria Belleudi, Francesca Romana Poggi.

**Investigation:** Marco Massari, Stefania Spila Alegiani, Francesca Menniti Ippolito.

**Methodology:** Marco Massari, Stefania Spila Alegiani, Cristina Morciano, Matteo Spuri, Pasquale Marchione, Patrizia Felicetti, Valeria Belleudi, Francesca Romana Poggi, Marco Lazzeretti, Michele Ercolanoni, Elena Clagnan, Emanuela Bovo, Gianluca Trifirò, Roberto Da Cas, Francesca Menniti Ippolito.

**Project administration:** Loriana Tartaglia, Francesca Menniti Ippolito.

**Software:** Marco Massari, Stefania Spila Alegiani, Matteo Spuri, Valeria Belleudi, Francesca Romana Poggi, Marco Lazzeretti, Michele Ercolanoni, Elena Clagnan, Emanuela Bovo.

**Supervision:** Marco Massari, Stefania Spila Alegiani, Cristina Morciano, Gianluca Trifirò, Ugo Moretti, Giuseppe Monaco, Olivia Leoni, Roberto Da Cas, Fiorella Petronzelli, Nadia Mores, Sarah Samez, Cristina Zappetti, Anna Rosa Marra, Francesca Menniti Ippolito.

**Validation:** Marco Massari, Stefania Spila Alegiani, Pasquale Marchione, Patrizia Felicetti, Valeria Belleudi, Francesca Romana Poggi, Marco Lazzeretti, Michele Ercolanoni, Elena Clagnan, Emanuela Bovo, Giovanna Zanoni, Paola Rossi.

**Writing – original draft:** Marco Massari, Stefania Spila Alegiani, Cristina Morciano, Francesca Menniti Ippolito.

**Writing – review & editing:** Marco Massari, Stefania Spila Alegiani, Cristina Morciano, Matteo Spuri, Pasquale Marchione, Patrizia Felicetti, Valeria Belleudi, Francesca Romana Poggi, Marco Lazzeretti, Michele Ercolanoni, Elena Clagnan, Emanuela Bovo, Gianluca Trifirò, Ugo Moretti, Giuseppe Monaco, Olivia Leoni, Roberto Da Cas, Fiorella Petronzelli, Loriana Tartaglia, Nadia Mores, Giovanna Zanoni, Paola Rossi, Sarah Samez, Cristina Zappetti, Anna Rosa Marra, Francesca Menniti Ippolito.

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
