## [Editor Report · Decision Letter 0]

8 Mar 2022

Dear Dr Morciano, 

Thank you for submitting your manuscript entitled "Post-marketing active surveillance of myocarditis and pericarditis following vaccination with COVID-19 mRNA vaccines in persons aged 12-39 years in Italy: a multi-database, self-controlled case series study" for consideration by PLOS Medicine.

Your manuscript has now been evaluated by the PLOS Medicine editorial staff and I am writing to let you know that we would like to send your submission out for external peer review.

Please re-submit your manuscript within two working days, i.e. by Mar 10 2022 11:59PM.

Kind regards,

Callam Davidson

Associate Editor

PLOS Medicine

---

## [Decision Letter · Decision Letter 1]

26 Apr 2022

Dear Dr. Morciano,

Thank you very much for submitting your manuscript "Post-marketing active surveillance of myocarditis and pericarditis following vaccination with COVID-19 mRNA vaccines in persons aged 12-39 years in Italy: a multi-database, self-controlled case series study" (PMEDICINE-D-22-00715R1) for consideration at PLOS Medicine. 

Your paper was evaluated by an associate editor and discussed among all the editors here. It was also discussed with an academic editor with relevant expertise, and sent to independent reviewers, including a statistical reviewer. The reviews are appended at the bottom of this email and any accompanying reviewer attachments can be seen via the link below:

[LINK]

In light of these reviews, I am afraid that we will not be able to accept the manuscript for publication in the journal in its current form, but we would like to consider a revised version that addresses the reviewers' and editors' comments. Obviously we cannot make any decision about publication until we have seen the revised manuscript and your response, and we plan to seek re-review by one or more of the reviewers. 

We expect to receive your revised manuscript by May 17 2022 11:59PM. Please email us (plosmedicine@plos.org) if you have any questions or concerns.

We look forward to receiving your revised manuscript. 

Sincerely,

Callam Davidson, 

Associate Editor

PLOS Medicine

plosmedicine.org

The Data Availability Statement (DAS) requires revision. If the data are not freely available, please describe briefly the ethical, legal, or contractual restriction that prevents you from sharing it. Please also include an appropriate contact (web or email address) for inquiries (this cannot be a study author).

Throughout, please consider using an alternative term instead of ‘brand’ to reflect the non-proprietary presentation of the vaccines e.g. product/type.

Abstract Methods and Findings:

* Please ensure that all numbers presented in the abstract are present and identical to numbers presented in the main manuscript text.

* Please include the important dependent variables that are adjusted for in the analyses.

Citations should be in square brackets and precede punctuation throughout.

Line 105: Please include ‘to our knowledge’ to temper claims of primacy.

The terms gender and sex are not interchangeable (as discussed in http://www.who.int/gender/whatisgender/en/ ); please use the appropriate term.

Please confirm whether informed consent from participants was written or verbal.

I could not locate the RECORD-PE checklist? Please check this was included and cite in your Methods (S1 Checklist).

"Did your study have a prospective protocol or analysis plan? Please state this (either way) early in the Methods section.

Please include the unadjusted comparisons as well as the adjusted comparisons in Tables 3 and 4 (if space is an issue, the unadjusted comparisons can be placed in the Supporting Information and cited in the Results).

References: Please use et al only after listing the first six authors.

Please update figures to use non-proprietary rather than commercial vaccine names, for consistency with the main text.

Comments from the reviewers:

Reviewer #1: See attachment

Michael Dewey

Reviewer #2: Myocarditis/pericarditis in relation to mRNA vaccinations has been a topic of interest lately due to ongoing concerns of an association. This study, along with others based in different countries, found significant associations between mRNA vaccinations and this event. This paper in particular finds an association [0-7) days after vaccination for mRNA-1273 following both doses and BNT162b2 following the second dose. This study is of interest as it supports findings found in other countries but uses self-controlled case series adapted to handle event-dependent exposures. This is a well-written study, but I have a few suggestions that I think would improve the paper and that I ask the authors to consider:

1. An important assumption in this study is that there are event-dependent exposures. The authors mention that they conduct a subanalysis using SCCS standard, starting the observation time at the first and second dose. I assume that this means that all time before vaccination is not considered in the SCCS analysis, and the days following the 21-day risk period are the reference period. While I agree that there could be bias introduced due to a delayed vaccination following an event, I do not think that removing time prior to vaccination in the standard SCCS method and comparing it to your main results is a sufficient example in itself of why the modified SCCS model should be used. I would recommend doing a plot similar to Figure 1, but include time before the first vaccination as well so that it can be investigated whether there is a visible "dip" in events just prior to vaccination. If so, introducing a pre-risk period is justified. Long event delays or preclusions of the exposure will not be visible on this plot, but doing the standard SCCS analysis without the exclusion of this time as a sort of sensitivity analysis to see whether the results are heavily biased by the event-dependent exposure assumption could aid in this investigation. If the event delays or precludes the exposure, then the relative incidence will be biased upwards. Alternatively, (Farrington CP, Whitaker H, Weldeselassie YG. Self-Controlled Case Series Studies. A Modelling Guide with R. CRC Press, 2018) recommends removing post-event exposures in a standard SCCS analysis in order to display this bias (pg. 237-239).

2. I do not follow the conclusion "According to our results the risk increased after the second dose and in the youngest for both vaccines, remained moderate following vaccination with BNT162b2, while was higher in males following vaccination with mRNA-1273." 

a. The youngest age group (12-17) did not have any estimates for mRNA-1273, so how can the risk be increased in the youngest for both vaccines? 

b. I would also use the phrase "was not statistically significant" instead of "moderate" when discussing RI. 

c. In general, this sentence is a bit hard to follow. Consider breaking it up into two.

3. As you reference later in the paper, different countries found varying results per dose when receiving the different vaccine brands. Therefore, I would be cautious assuming that heterologous vaccinations essentially do not exist in the primary analysis. Is this a reasonable assumption to make following the Italian vaccination plan that many do not get cross-vaccinated? I would report the results of the analysis where these individuals have been censored when they received a different vaccine brand as the main analysis to avoid the effect mixing the brands would have.

4. This comment is regarding Table S12: Sensitivity Analysis: c) Heterologous vaccination (II). If it is correct that only one person's time is censored here (and it is a person who got an mRNA-1273 to BNT162b2 combination) and the rest of the analysis remains the same, why does this change the number of cases in BNT162b2 such that it is no longer estimable? Should it not be the same number of cases? Please recheck this analysis or explain the analysis in more detail. 

5. Using the time after the exposure period for a reference period allows for more cases and power to your study instead of just using the time prior to vaccination as the reference period. However, performing a sensitivity analysis on the length of the exposure period is, therefore, more important as it affects your reference period. For that I have two comments:

a. Have you preformed any sensitivity analyses using this risk period (either shortening or extending the length) justifying your choice of risk period? Other countries listed in your discussion have a risk period of 28-days.

b. I would recommend adding a sensitivity analysis either where you increase the number of days in your risk period or exclude some time shortly following your risk period so that you can investigate whether your unexposed period remains unbiased from potential delayed reported events related to your exposure. 

6. A schematic illustration of the time windows used in the SCCS analysis would benefit the reader. This could potentially be put in as a supplementary plot and referenced to in "Statistical analysis" or "Definition of exposures".

Minor comments:

* Line 186: SCSS -> SCCS

* Table 1: Please format the table so that the groupings of characteristics are more visible and the group-wise percentages are easier to see.

* Fig 1: Please change the heading of the figure to BNT162b2 and mRNA-1273 to match the rest of the paper. 

* Lines 317-318: The statement in these lines is correct for mRNA-1273, but only after the second dose for Pfizer. Consider rephrasing.

* Line 338: "also reported an higher risk in male" -> "also reported a higher risk in males"

* Lines 355-356: This sentence does not make sense. Consider rephrasing. "The large sample size (about 3 million vaccinated people aged 12-39 years) allowed us to look at fine risk intervals following vaccination and conduct several subgroup analyses."

* Lines 380-383: This sentence does not make sense. Consider rephrasing.

* Line 389: Remove "s" in "conclusions".

Reviewer #3: Massari et al. estimate the increased risk for myocarditis and pericarditis following receipt of mRNA Covid-19 vaccines. The study is interesting, but I do have concerns that merit addressing.

Note: The system describes this as a revision. I was not involved in the original submission and know nothing of previous comments. I therefore review this as I would review the original submission.

== Major ==

More needs to be said about the quality of the data, particularly as it synthesizes various sources. Are all sources guaranteed to be complete? If not, why and how incomplete? How long is the history for each person? How complete is the follow-up? Etc.

In the introduction, the authors should explicitly state the scientific gap they aim to feel. From what I understand, the relative risk has already been quantified for these two vaccines, including by age group and sex. What then is the goal of the study?

I do not have intimate knowledge of the SCCS methodology, and particularly not of its extension to situations in which exposure depends on past events (like it does here, as past myocarditis lowers the probability of vaccination). This is a critical aspect of this study, and others more knowledgeable should be sought to review it, as many questions arise including issues of confounding by calendar time, inclusion only of vaccinated (e.g., why isn't information on the unvaccinated used? Does this not result in bias?), etc.

In the same context, While I hesitate to suggest a different analysis and I understand the within-person advantage of SCCS, but considering that other than previous myocarditis, there aren't known strong confounders for the vaccination => myocarditis effect, why not use a standard cohort design with a modified Poisson model? This is even more warranted when considering that the authors acknowledge that not including unvaccinated persons could lead to bias.

The authors should avoid language that stems from statistical significance. E.g., saying that no association was found between the first 7 days after the first dose of the Pfizer vaccine and the outcome or that "increased risk was present only in…". The reason is simple - if statistical significance is evoked, then all its tenets must be adhered to, including adjusting for multiple comparisons, which was not performed here. A simple reporting of the CI, then, would suffice.

== Minor ==

The abstract result section is overly detailed. Should only be overall effects per each vaccine and maybe some specific interesting subgroup results.

The conclusion in the abstract "This population-based study suggested that mRNA vaccines were associated with myocarditis/pericarditis in the population younger than 40 years, whereas no association was observed in older subjects" is weird, considering that this study is limited to ages 12-39. Later I see that this was a secondary analysis. Its results then, should not appear in the abstract.

Persons with missing data were excluded. Why? How many were there? A population flow chart should be included and the number of missing mentioned.

The authors dedicate Table 1 to all those vaccinated with mRNA vaccines. This is not warranted. The study population in a SCCS design are those that experienced an event and were included. The complete population should only be mentioned in the context of the population flow chart (which I do not see in this submission). Similarly, mentioning the full population as a strength does not seem to make sense.

The structure of Table 2 is not optimal. The stratification by age makes the table hard to understand and does not seem helpful.

The effect estimate of interest is called "relative incidence" throughout. It would be preferable to use the more informative and accurate "incidence rate ratio".

The strengths section is better removed. The merit of a study should be evident from the reporting and not specifically mentioned. Moreover, most of this section is devoted to what is actually a limitation - the exclusion of unvaccinated individuals.

Reviewer #4: See attached file

[LINK]

---

## [Decision Letter · Decision Letter 2]

15 Jun 2022

Dear Dr. Morciano,

Thank you very much for re-submitting your manuscript "Post-marketing active surveillance of myocarditis and pericarditis following vaccination with COVID-19 mRNA vaccines in persons aged 12-39 years in Italy: a multi-database, self-controlled case series study" (PMEDICINE-D-22-00715R2) for review by PLOS Medicine.

I have discussed the paper with my colleagues and the academic editor and it was also seen again by three reviewers. I am pleased to say that provided the remaining editorial and production issues are dealt with we are planning to accept the paper for publication in the journal.

The remaining issues that need to be addressed are listed at the end of this email. Please take these into account before resubmitting your manuscript:

We look forward to receiving the revised manuscript by Jun 22 2022 11:59PM.   

Sincerely,

Callam Davidson, 

Associate Editor 

PLOS Medicine

plosmedicine.org

Requests from Editors:

Lines 60-61: Please update to ‘The main study limitations were that the outcome was not validated through review of clinical records, and there was an absence of information on the length of hospitalization and, thus, the severity of the outcome.’

Line 163: Please specify the variables that were considered relevant.

Line 195: Please update to ‘we extended the exposure period to 28 days as well reducing it to 14 days.’

Line 200: Please describe these further sensitivity analyses.

Line 204: Please update to ‘This study is reported as per the REporting of studies Conducted using Observational Routinely-collected Data for Pharmacoepidemiology (RECORD-PE) checklist (S1 Checklist).’

Please update your RECORD-PE checklist to use section names and paragraph numbers rather than page/line numbers (these will either change or not be present in the final published version).

Line 215: Please include the actual number of missing observations as well as the proportion.

Line 269: Please update to ‘Of note, there were…’

Line 280-281: Please include the 95% CIs. 

Line 386: Please update to ‘It is reassuring that the sensitivity analysis (S17 Table) in which the observation time began at first or second vaccination dose in a standard SCCS method gave similar estimates to our main analysis.’

Line 391: ‘Such bias is probably minimal’ – please either provide a reference to support this claim or remove it.

For Internet references (e.g. reference 8), please include the data accessed. 

Please ensure all Supporting Information items (e.g. Supporting Tables/Figures) are cited in the main text. 

To help us extend the reach of your research, please provide any Twitter handle(s) that would be appropriate to tag, including your own, your coauthors’, your institution, funder, or lab. Please respond to this email with any handles you wish to be included when we tweet this paper.

Comments from Reviewers:

Reviewer #1: The authors have addressed all my points.

Michael Dewey

Reviewer #2: Thank you for including my suggested changes and analyses to the article. 

I have no further comments. I recommend this article be accepted.

Reviewer #3: The bulk of my comments were answered in a satisfactory manner.

My most important comment was regarding the need for expert evaluation of the SCCS methodology. It seems that this was now done by a key developer of said methodology.

I have nothing further to add.

Because in my opinion, the decision for this paper should hinge mostly on methodological considerations, and because I am not an expert in said methodology, I defer to the recommendation of more knowledgeable reviewers.

---

## [Editor Report · Decision Letter 3]

21 Jun 2022

Dear Dr. Morciano,

Thank you very much for re-submitting your manuscript "Post-marketing active surveillance of myocarditis and pericarditis following vaccination with COVID-19 mRNA vaccines in persons aged 12-39 years in Italy: a multi-database, self-controlled case series study" (PMEDICINE-D-22-00715R3) for review by PLOS Medicine.

There are a few minor remaining issues that need to be addressed before we can proceed to acceptance - these are listed at the end of this email. Please take these into account before resubmitting your manuscript.

As before, when revising the manuscript, please ensure you address the specific points made by each reviewer and the editors. In your rebuttal letter you should indicate your response to the reviewers' and editors' comments and the changes you have made in the manuscript. Please submit a clean version of the paper as the main article file. A version with changes marked must also be uploaded as a marked up manuscript file.

Given the relatively minimal comments, we hope to receive your revised manuscript within 2 days. Please email me(cdavidson@plos.org) if you foresee any problems with this deadline. Otherwise, we look forward to receiving the revised manuscript by Jun 23 2022 11:59PM.   

Sincerely,

Callam Davidson, 

Associate Editor 

PLOS Medicine

plosmedicine.org

Requests from Editors:

Thank you for providing your Author Summary. I have reviewed and suggested updated wording below. Please carefully review this wording and confirm that it is a) scientifically accurate and b) presents the key findings in a manner that you deem appropriate. Feel free to edit if you feel anything could be clearer.

Author summary

Why was this study done?

* Pharmacovigilance reports and observational studies have suggested an increased risk of myocarditis/pericarditis following the COVID-19 mRNA vaccine administration in people younger than 40 years.

* More information on the safety of COVID-19 mRNA vaccines is needed to further explore the relationship between mRNA vaccines and myocarditis/pericarditis in this population.

What did the researchers do and find?

* We conducted a multiregional self-controlled case series study in Italy between 27 December 2020 - 30 September 2021 to investigate the association between myocarditis/pericarditis and COVID-19 mRNA vaccines in the population aged 12-39 years (n=2,861,809). 

* We found 441 myocarditis/pericarditis cases, 114 of which occurred within the 21-day risk interval after vaccination. 

* Within the 21-day risk interval, the relative incidence (RI) was 1.99 (95% confidence interval [CI] 1.30-3.05) after the second dose of BNT162b2 and 2.22 (1.00-4.91) and 2.63 (1.21-5.71) after the first and second doses of mRNA-1273, respectively. Within the 0–7-day risk interval, the RI was 6.55 (2.73-15.72) after first dose of mRNA-1273, and 3.39 (2.02-5.68) and 7.59 (3.26-17.65) after the second doses of BNT162b2 and mRNA-1273, respectively.

* The highest risk was seen in males, 0-7 days after the first and second dose of mRNA-1273 (RIs of 12.28 (4.09-36.83) and 11.91 (3.88-36.53), respectively). After the second dose of mRNA-1273 in males, the excess of cases (EC) was 8.8 (4.9-12.9) per 100,000 vaccinated individuals.

What do these findings mean?

* Consistent with previous studies, the findings suggest that COVID-19 mRNA vaccines were associated with myocarditis/pericarditis in the population younger than 40 years.

* The results provide information that could be helpful for the continuous assessment of the post-marketing benefit/risk profile of the COVID-19 mRNA vaccines and should be considered within the context of the proven mRNA vaccine effectiveness in reducing COVID-19 morbidity and mortality.

Line 64: Please update 'According to our results the risk increased after second dose of BNT162b2 and after both doses of mRNA-1273' to 'According to our results, increased risk of myocarditis/pericarditis was associated with the second dose of BNT162b2 and both doses of mRNA-1273'.

Line 67 (abstract conclusions): Please change 'weighed' to 'considered'.

---

## [Editor Report · Decision Letter 4]

22 Jun 2022

Dear Dr Morciano, 

On behalf of my colleagues and the Academic Editor, Dr James Beeson, I am pleased to inform you that we have agreed to publish your manuscript "Post-marketing active surveillance of myocarditis and pericarditis following vaccination with COVID-19 mRNA vaccines in persons aged 12-39 years in Italy: a multi-database, self-controlled case series study" (PMEDICINE-D-22-00715R4) in PLOS Medicine.

PRESS

Sincerely, 

Callam Davidson 

Associate Editor 

PLOS Medicine